# Unveiling post-vaccination proteomic signatures in SARS-CoV-2 infection-naïve individuals associated with Omicron breakthrough infections

Yiwen Liu[1⊙], Eric Lu[2⊙], Katherine D. Ellingson[1], James Hollister[1], Tuo Liu[3], Wadana Hamzazai[1], Shawn C. Beitel[3], Alberto J. Caban-Martinez[4], Manjusha Gaglani[5], Allison L. Naleway[6], Lauren E. W. Olsho[7], Andrew L. Phillips[8], Natasha Schaefer Solle[9], Harmony L. Tyner[10], Sarang K. Yoon[8], Karen Lutrick[11], Jefferey L. Burgess[3]*

**1** Department of Epidemiology and Biostatistics, Mel and Enid Zuckerman College of Public Health, University of Arizona, Tucson, Arizona, United States of America, **2** Department of Biological Engineering, Massachusetts Institute of Technology, Cambridge, Massachusetts, United States of America, **3** Department of Community, Environment and Policy, Mel and Enid Zuckerman College of Public Health, University of Arizona, Tucson, Arizona, United States of America, **4** Department of Public Health Sciences, University of Miami, Miller School of Medicine, Miami, Florida, United States of America, **5** Baylor Scott and White Health, Temple, Texas and Baylor College of Medicine, Temple Texas, United States of America, **6** Kaiser Permanente Center for Health Research, Portland, Oregon, United States of America, **7** Abt Global LLC, Rockville, Maryland, United States of America, **8** Division of Occupational and Environmental Health, Spencer Eccles Fox School of Medicine, Rocky Mountain Center for Occupational and Environmental Health, Salt Lake City, Utah, United States of America, **9** Department of Medicine, University of Miami, Miller School of Medicine, Miami, Florida, United States of America, **10** Dartmouth Hitchcock Medical Center, Lebanon, New Hampshire, United States of America, **11** Department of Family and Community Medicine, College of Medicine – Tucson, University of Arizona, Tucson, Arizona, United States of America

⊙ These authors contributed equally to this work.
* jburgess@email.arizona.edu

## Abstract

### Background

Given the persistence of the SARS-CoV-2 virus, it is important to understand the proteome associated with breakthrough infections among COVID-19 vaccinated individuals.

### Methods

We conducted a nested case-control study within the frontline worker HEROES-RECOVER cohorts to specify a study population of SARS-CoV-2 infection-naïve participants who had a third dose of COVID-19 origin strain WA-1 monovalent mRNA vaccine from August 2021 to January 2022. We compared serum proteomic profiles for those who subsequently experienced Omicron breakthrough infections with those of matched controls without infections. Our study leveraged proteomics data generated from the SomaScan Platform and adopted a robust feature selection method, elastic net regularized conditional logistic regression with bootstrapping, to

**Data availability statement:** Data cannot be shared publicly by the authors because it is owned by the Centers for Disease Control and Prevention (CDC), and the data contains personal identifying information. Data are available upon request pending approval from the CDC for researchers who meet the criteria for access to confidential data. Requests for de-identified data can be sent to LTO7@cdc.gov.

**Funding:** National Center for Immunization and Respiratory Diseases, Centers for Disease Control and Prevention under contract numbers 75D30120R68013 awarded to Marshfield Clinic Research Laboratory and 75D30120C08379 to University of Arizona. The funder's had no role in study design, data collection and analysis, decision to publish, or preparation of this proteomics manuscript.

**Competing interests:** The authors have declared that no competing interests exist.

identify key proteins. Enrichment analyses were performed to investigate biological pathways.

## Results

We identified 28 significant proteins out of over 7,000 candidate proteins. Key findings included downregulated chemokines (CXCL2, CXCL3, CCL19, CCL23) and elevated cytokine IL-7 levels in breakthrough cases, with pathway analysis revealing enrichment in chemokine signaling and cytokine-cytokine interaction pathways. Other key proteins, such as LGALS1, HAVCR2, and SELE were upregulated in breakthrough cases.

## Discussion

These results reveal potential immune response mechanisms in breakthrough infections, characterized by viral immune evasion and compensatory T-cell regeneration. The identified biomarkers may provide valuable insights for future predictive profiles and therapeutic strategies.

## Introduction

The severe acute respiratory syndrome coronavirus 2 (SARS-CoV-2) has caused infections in hundreds of millions of people worldwide and is likely to remain in circulation following endemic seasonal patterns [1,2]. Although widespread vaccination has significantly reduced the severity of outbreaks, the virus remains persistent due to its ability to evolve and escape from vaccine-mediated immune protection. For this reason, breakthrough infections remain a concern, suggesting the importance of studying the proteomic differences across fully vaccinated individuals with and without breakthrough infections. Most existing studies of breakthrough infection have focused primarily on antibody titers, neutralizing capacity, or antigen-specific T-cell response. Although this approach is critical for understanding immunologic response as a mechanism behind breakthrough infection, they capture a subset of the complex host response to infection following vaccination. Proteomic profiling provides a comprehensive and integrative view of the plasma protein signatures associated with breakthrough infection by quantifying thousands of proteins simultaneously. Examining the larger proteome can generate insights into the mechanism through which the immune system interacts with the virus post-vaccination, identify predictive biomarkers and significant pathways of breakthrough infections, and potentially shed light on the development of therapeutic interventions to prevent breakthrough infections [3–7].

Previous studies on SARS-CoV-2 breakthrough infection have focused on a limited number of candidate proteins. Zhang et al. studied the proteome features of patients who received a third vaccination, together with those experiencing Omicron breakthrough infections [8]. Their findings revealed an upregulated chemokine signaling

pathway and humoral response markers (IgG2 and IgG3) in breakthrough infections compared to vaccination-induced immunity. Kawasuji et al. investigated acute immune responses in both vaccinated and unvaccinated patients during Omicron infection [9]. They found that high IL-6 levels correlated with strong neutralization antibody response. Although these studies offer valuable insights into the breakthrough cases, they are limited to targeted or hypothesis-driven proteins, limiting the ability to capture a broader range of biological pathways that may distinguish individuals who experienced breakthrough infections from those who remain uninfected despite similar vaccination histories.

The HEROES-RECOVER prospective cohort study [10,11] offers a unique opportunity to investigate the proteomic landscape of Omicron breakthrough infections within an infection-naïve population; the analysis focuses on samples collected during the early stage of Omicron variant predominance. In contrast to data generated from recent cohorts with varied infection and vaccination histories, this dataset provides a controlled and well-characterized group of individuals with fewer confounding factors. Therefore, our study addresses an essential research gap in studying the proteomic landscape of breakthrough infection by focusing on a well-defined, infection-naïve population.

We hypothesized that vaccinated individuals who experienced Omicron breakthrough infections may exhibit distinct proteomic signatures, reflecting different immune activation mechanisms, inflammatory regulation, and host-response pathways. To better understand the molecular mechanisms underlying SARS-CoV-2 Omicron breakthrough infections in vaccinated individuals, we investigated more than 7,000 candidate plasma protein profiles obtained from Omicron breakthrough infection cases and matched uninfected control samples. The proteomics data were processed through the SomaScan Platform [12–14], which allows the simultaneous quantification of thousands of proteins in each sample. The objective of our investigation was to uncover significant protein markers and biological pathways associated with breakthrough infection cases, which could yield insight into the mechanisms driving breakthrough infections in vaccinated individuals.

## Methods

### Study design

A nested case-control study was designed within a large prospective cohort of frontline workers from eight locations in the US during early Omicron predominance (defined here as December 2021 through September 2022). Breakthrough cases were defined as those with a first-time SARS-CoV-2 infection at least 14 days following a third dose of COVID-19 origin strain WA-1 monovalent mRNA vaccine, and controls were defined with the same criteria but without a breakthrough infection.

### Participants

Beginning in July 2020, frontline workers were followed in prospective cohorts through the Arizona Healthcare, Emergency Response, and Other Essential workers Study (HEROES) and the Research on the Epidemiology of SARS-CoV-2 in Essential Response Personnel (RECOVER) sites in Arizona, Florida, Minnesota, Oregon, Texas, and Utah [10,11]. Briefly, eligible participants included adults who worked at least 20 hours per week in occupations requiring frequent direct contact with non-household members (i.e., healthcare workers, first responders, and other public-facing frontline workers). Upon enrollment, participants completed a survey on sociodemographic characteristics, occupation, health status, health-related behaviors, and prior SARS-CoV-2 infection. COVID-19 vaccination information was obtained through surveys that were regularly sent to participants when they became eligible for vaccination according to evolving vaccine guidelines and validated through electronic health records or vaccine registries. Additional surveys were conducted upon infection, and information on mask use and exposures was collected quarterly, upon infection, or upon onset of illness symptoms.

Participants provided a mid-turbinate nasal specimen weekly and with onset of any illness symptoms. These specimens were tested for SARS-CoV-2 using reverse transcription-polymerase chain reaction (RT-PCR) at the Marshfield Clinic Laboratory (Marshfield, WI). Additionally, blood samples were collected at the following time points: (1) upon enrollment;

(2) quarterly; and (3) approximately 14–60 days after any immunity-conferring event, such as SARS-CoV-2 infection or COVID-19 vaccination. All study protocols were reviewed and approved by the Institutional Review Boards at each site, and participants provided informed consent for all study activities.

### Eligibility criteria and case definition

Omicron breakthrough infections were defined as SARS-CoV-2 infections caused by the Omicron variant occurring at least 14 days following a third dose of COVID-19 mRNA vaccine in individuals with no history of previous infection. In-study infections were determined using the self-collected mid-turbinate sample provided weekly and with any onset of symptoms. Prior infection history was ascertained via self-report upon study enrollment or a negative qualitative antibody result from the baseline blood draw using a locally-developed and validated semi-quantitative enzyme-linked immuno-sorbent assay (ELISA), which measured antibody binding to the receptor binding domain (RBD) and S2 subunit domain (S2) of the SARS-CoV-2 Washington-1 spike protein, as previously described [15]. Variant of infection was confirmed by whole-genome sequencing of eligible specimens or estimated based on the state-specific predominant variant at the time of infection according to Centers for Disease Control and Prevention data [16].

Participants were eligible for inclusion if they had received three doses of an mRNA COVID-19 vaccine, did not withdraw from HEROES/RECOVER prior to Omicron becoming the site-specific dominant variant, had a blood draw at any time after the third vaccine dose but prior to any additional COVID-19 vaccine or any SARS-CoV-2 infection, and had no history SARS-CoV-2 infection at the time of the blood draw. Among those included individuals, a case was then defined as a participant who tested positive for SARS-CoV-2 infection after the blood draw and prior to any additional vaccine dose (Fig 1).

### Matching

Subjects were matched into pairs of case and control individuals in a 1:1 ratio using a greedy matching algorithm based on the following criteria: study site (Tucson, AZ; Phoenix, AZ; Other location, AZ; Temple, TX; Portland, OR; Duluth, MN; Salt Lake City, UT), sex, race/ethnicity (non-Hispanic/White; non-Hispanic/Black; non-Hispanic/Asian; Hispanic; other race), number of chronic conditions (0, 1, 2, or 3 + chronic conditions), age (±3 years), and time between third vaccine dose and blood draw. If no exact match was found, then the case was excluded from the analysis. Self-reported chronic conditions included asthma, chronic lung disease, cancer, diabetes, heart disease, hypertension, immunosuppression, kidney disease, liver disease, neurologic or neuromuscular disease, and autoimmune disease. To ensure a close match on the time between a third vaccine dose and blood draw, the control selected was the one with the smallest difference in the number of days between the vaccination and blood draw dates as compared to the case. For case individuals in which the blood draw occurred less than 150 days after vaccination, this difference must have been less than 21 days. Although immunosuppressed individuals were a minority population, the Fisher's exact test was used to ensure immunosuppression was not overrepresented in either case or control groups and would not significantly confound findings related to immune activity between the experimental groups.

### Sample preprocessing and proteomics

Blood samples were collected in 8.5 mL Vacutainer tiger-top serum separator tubes (SST) and allowed to clot for at least an hour at room temperature. Samples were then centrifuged at 1300 rpm for 15 minutes at 4°C and stored at 4°C for up to 24 hours on weekdays or at −20°C on weekends before receipt at the University of Arizona or other site research laboratories for preprocessing. After receipt, if frozen, the sample was thawed at room temperature for 1 hour, and the serum was divided into 1.8 mL aliquots. Individual aliquots were stored at −80°C until further analysis. Serum aliquots were sent to SomaScan Assay platform [12] for proteomics profiling in 7k format. Relative abundance, measured in relative fluorescent units (RFU), was returned for each SOMAmer (tentatively annotated protein) per sample.

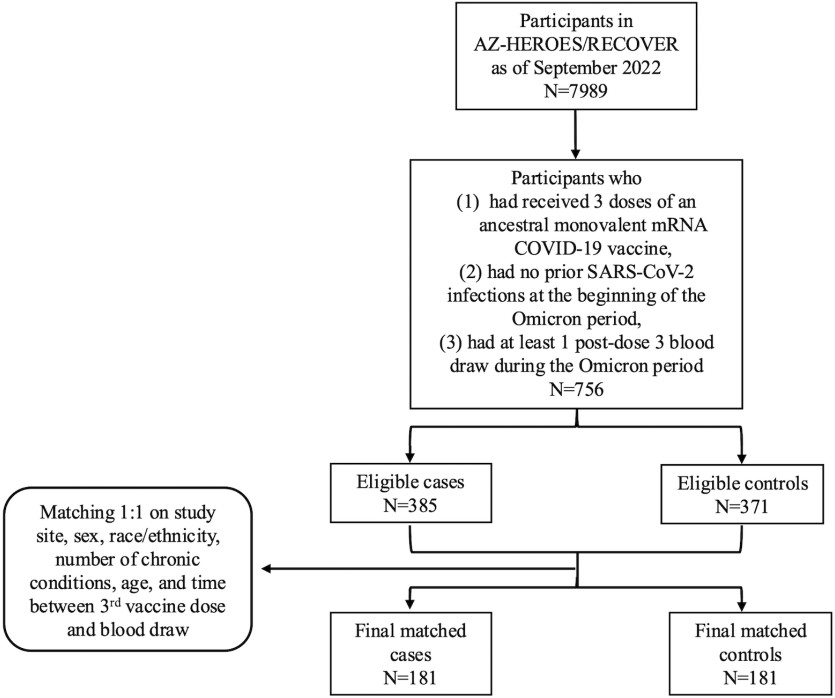

**Fig 1. Flowchart showing inclusion and exclusion criteria for nested case-control studies within the HEROES/RECOVER cohort of frontline workers: 181 cases of first-time post-vaccination Omicron infection following a third dose of COVID-19 origin strain WA-1 monovalent mRNA vaccine) were matched 1:1 to controls based on study site, sex, race/ethnicity, number of chronic conditions, age, and time between 3rd vaccine dose and blood draw.**

## Data preprocessing and quality control

To remove assay and sample bias from proteomics signals, the standard SomaScan v4.1 data standardization pipeline was applied. The steps are as follows: 1) "hybridization control normalization" using twelve hybridization control sequences added to each sample to adjust for biases arising from hybridization, 2) "intraplate median signal normalization" using all SOMAmer reagents on a sample to remove systematic biases between samples within a plate, 3) "plate scaling" using a pooled calibrator sample across plates of identical sample matrices to adjust for inter-plate variation, 4) "calibration" using a pooled calibrator to correct for SOMAmer-level variation, and 5) "adaptive normalization to a reference" using a healthy cohort's SOMAmer RFUs as reference to standardize each sample's overall signal. A final check is performed with replicate quality control samples that were run alongside clinical samples to assess the accuracy of SOMAmer reagents' signal compared to the quality control samples.

The acceptance criteria for steps 1–3 require scale factors to be between 0.4 and 2.5, and between 0.8 and 1.2 for step 4. Step 5 required that at least 30% of the sample value lies between two standard deviations of the healthy population reference.

## Statistical modeling

We utilized the Wilcoxon signed-rank test to evaluate the between-group differences for the number of chronic conditions, Fisher's exact test for the number of immunosuppressed versus non-immunosuppressed individuals, and two-sample t-test for other continuous characteristics, including age, days from third dose to case infection, and average hours exposed to COVID per week. Necessary transformations were applied to ensure the normality of the data distribution.

Given the high-dimensional nature of the proteomics data, a robust feature selection algorithm was needed to identify key proteins of interest and reduce the risk of model overfitting. Further, to handle the matched pair structure and the binary response variable, we chose to utilize elastic net regularized conditional logistic regression, hereon referred to as EN-CLR, as our model. In general, penalized regression has the property of "selecting" important features by reducing the coefficients of less relevant features to zero. We opted to use the elastic net penalty instead of the LASSO or Ridge penalty as the LASSO penalty is liable to select only one covariate out of a set of highly correlated but equally "important" covariates, which is an undesirable property for identifying biologically important features, and the Ridge penalty tends to introduce algorithmic instability for the penalized conditional logistic regression [17]. The elastic net penalty has been shown to generate better performing models in assessing datasets with many correlated features compared to LASSO [18,19] – as is the case with our dataset (S1 Fig).

To ensure stability in our results, rather than directly applying the bootstrap method to the original dataset, we generated 1,000 bootstrap resampled datasets and applied EN-CLR to each bootstrap. We performed bootstrapping by resampling entire strata with replacement with random seeds 1–1000. The same seeds were used for downstream analyses involving nondeterministic outcomes. For each bootstrap dataset, the parameter determining the strength of regularization was fitted using 10-fold cross-validation, and the model with the lowest deviance was selected. Each resulting model contains a set of proteins with non-zero coefficients yielding a total of 1,000 sets of selected proteins. Additionally, we recorded the frequencies that each protein was selected and denoted the frequency set as.

The proposed method is closely related to the conventional stability selection as originally characterized by Meinshausen and Bühlmann [20] but differing in three regards: 1) elastic net penalty instead of LASSO penalty, 2) bootstrapping instead of subsampling, and 3) cross-validation to obtain the regularization parameter instead of predefining an array of, then selecting the value maximizing a predictor's selection probability. Meinshausen and Bühlmann demonstrated that bootstrapping and subsampling exhibited similar behaviors while the selection of was not typically a strong determinant of selection outcome.

All statistical analyses were performed using R version 4.2.2. Specifically, the elastic net regularized conditional logistic regression models were fit using the clogitL1 R package with the participant outcome (breakthrough infection/control) as the binary response variable and all proteins (7,289 total) as independent variables [17]. Unregularized conditional logistic regression models were fit using the survival R package with case (breakthrough infection) versus control status as the binary response variable and the $\log_2$ transformed relative fluorescence units of proteins resulting from stability selection (28 total) as independent variables. Each package provided cross-validation functionalities, and each sample remained with their stratum during cross-validation. Data for each protein was scaled to a mean of 0 and standard deviation of 1 prior to model fitting.

## Pathway analysis

To better understand the functional implications of the highly selected (HS) proteins, we utilized two methods of pathway analysis: 1) over-representation analysis (ORA) which leverages the hypergeometric distribution to assess how well a set of proteins of interest is represented in a pathway compared with random chance, and 2) Signaling Pathway Impact Analysis (SPIA) [21] which combines ORA with information about the total accumulated perturbation of protein abundance within pathways between groups, to identify pathways of importance.

We conducted overrepresentation analyses using the clusterProfiler R package [22], with all the available proteins as the background. P-values for significance of overrepresentation were false discovery rate (FDR)-adjusted for multiple comparisons to avoid false positives. For SPIA, we first calculated the $\log_2$ fold change of each protein. Because SPIA's internal KEGG pathways were outdated, we used KEGG's API to retrieve the most recent pathway versions and processed them with SPIA. SPIA reports and combines p-values for both pathway overrepresentation and the accumulated perturbation of protein abundance between conditions. This combined p-value is then FDR adjusted and reported in Results.

# Results

## Participant characteristics and matching

Prior to matching, 385 and 371 participants were eligible for the study as cases and controls, respectively. Their characteristics are summarized in Supplemental S3 Table. Following the matching process, a total of 362 participants (181 cases and controls) was included in the final matched dataset, and their characteristics are summarized in Table 1. The average age for cases and controls was 43 years (SD = 10 years). Most participants were non-Hispanic and white (97%), female (74%), and had no self-reported chronic conditions (70%). The average number of days between the third vaccine dose and blood draw was 28 days for cases and 26 days for controls. No significant differences were observed between case and control groups in any of the measured characteristics, including variables matched with tolerances (e.g., age and number of chronic conditions). The final match cohort remains representative of the broader eligible population.

**Table 1. Demographics and health characteristics for frontline workers participating in the study. Participants were matched on site, age, race/ethnicity, gender, number of chronic conditions, and days between third vaccine dose and blood draw. Matching differences in age and number of conditions are tolerated within ±3 years and ±1 condition, respectively.**

| | Cases (n = 181) | Controls (n = 181) | p-value |
|---|---|---|---|
| Site, n (%) | | | 1.00 |
| Tucson, AZ | 18 (9.94) | 18 (9.94) | |
| Phoenix, AZ | 17 (9.39) | 17 (9.39) | |
| Other areas in AZ | 9 (4.97) | 9 (4.97) | |
| Temple, TX | 3 (1.66) | 3 (1.66) | |
| Portland, OR | 20 (11.05) | 20 (11.05) | |
| Duluth, MN | 70 (39.23) | 70 (39.23) | |
| Salt Lake City, UT | 44 (24.31) | 44 (24.31) | |
| Age, mean (SD) | 43.43 (9.96) | 43.27 (9.88) | 0.88 |
| Race/ethnicity, n (%) | | | 1.00 |
| Non-Hispanic, white | 176 (97.24) | 176 (97.24) | |
| Non-Hispanic, Asian | 2 (1.10) | 2 (1.10) | |
| non-Hispanic/Black | 0 (0.00) | 0 (0.00) | |
| Hispanic | 2 (1.10) | 2 (1.10) | |
| Other | 1 (0.55) | 1 (0.55) | |
| Sex, n (%) | | | 1.00 |
| Male | 47 (25.97) | 47 (25.97) | |
| Female | 134 (74.03) | 134 (74.03) | |
| Chronic conditions, n (%) | | | 0.91 |
| 0 | 126 (69.61) | 126 (69.61) | |
| 1 | 25 (13.81) | 19 (10.50) | |
| 2 | 19 (10.50) | 27 (14.92) | |
| 3+ | 11 (6.08) | 9 (4.97) | |
| Days from third dose to blood draw, mean (SD) | 27.97 (25.97) | 26.02 (21.98) | 0.44 |
| Average hrs. exposed to COVID per week, mean (SD) | 4.87 (8.18) | 4.07 (7.80) | 0.55 |
| Immunosuppressed, n (%) | 7 (3.87) | 4 (2.21) | 0.54 |

## Protein markers for Omicron breakthrough infection cases

We applied the proposed EN-CLR method to the proteomics data. Out of the 7289 candidate proteins, 3113 were selected in at least one bootstrap model, reinforcing concerns of overreliance on a single model. We therefore only considered the proteins selected in the majority of the models (selection frequency) as our final set of proteins. These proteins, referred to as "highly selected proteins" (HS proteins), were used for downstream functional analyses (Fig 2A). A total of 28 proteins met this requirement and were subsequently used to fit a classical conditional logistic regression model to determine the expected difference in protein abundance between cases and controls (S1 Table). While multiple regression in high dimensions and penalized regression coefficient values do not easily lend themselves to interpretation, the directionality of HS proteins' coefficients was consistent across every bootstrap resample, and 10 of the HS proteins were found to be lower in abundance in case individuals compared with controls while 18 were found to be higher. Among those HS proteins, four belong to the chemokine family (CXCL2, CXCL3, CCL19, CCL23), which are essential for virus control and are directly involved in the COVID 19 infection [23]. Additionally, the T-cell development cytokine IL-7 showed higher serum levels in breakthrough infection cases.

We compared our results to a univariate method by selecting proteins of interest based on a threshold fold change and p-value derived from the paired two-sample t-test performed on each protein. While many proteins were nominally significant, adjusting for FDR suggested insufficient evidence to assign importance to any protein, confirming the need for more advanced feature selection methods (Fig 2B). The selection of HS proteins differed considerably from a threshold fold change and p-value approach, suggesting sensitivity to higher order relationships.

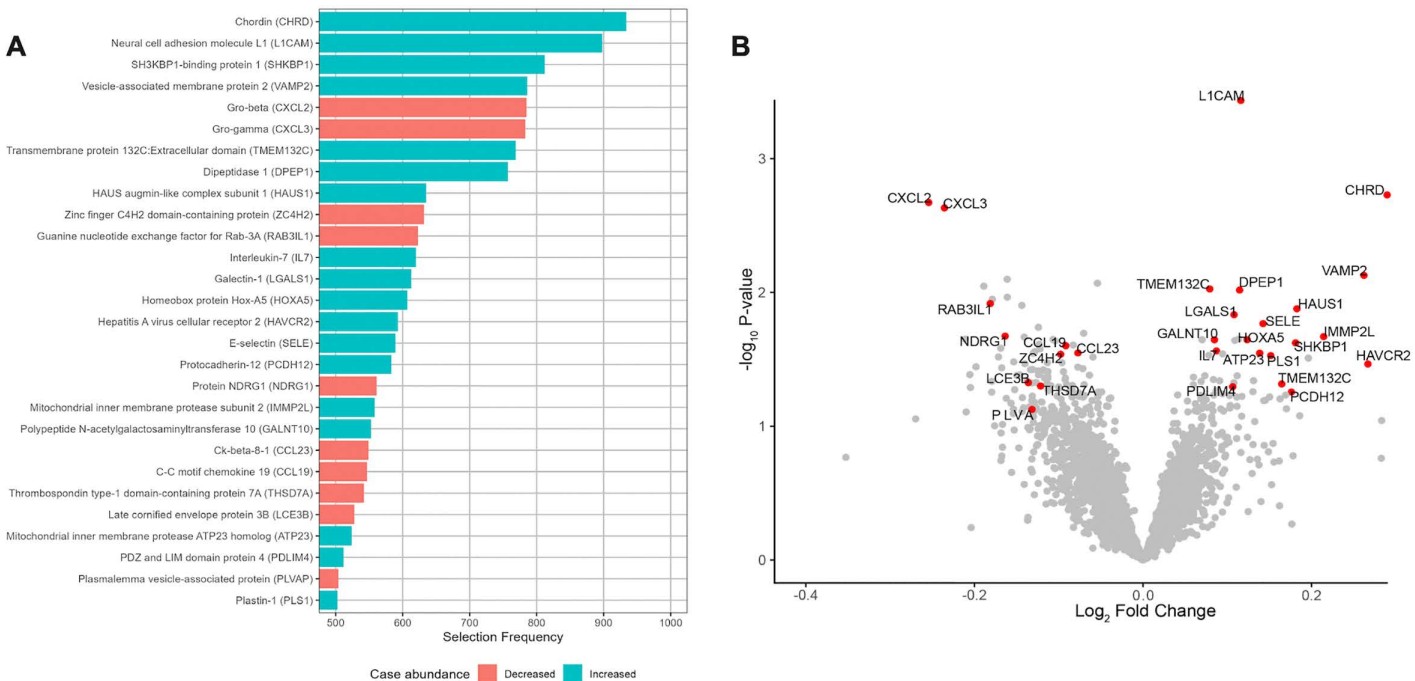

**Fig 2. EN-CLR on 1000x bootstrap resamples selects 28 proteins in a majority of models. A)** Red and blue bar colors indicate lower and higher abundance of protein in cases compared with controls, respectively. **B)** Volcano plot showing mean $\log_2$ effect sizes versus $-\log_{10}$ p-values of two-sample paired t-tests on log2 RFU between conditions. Highly selected proteins are labeled and shown in red.

## Enrichment of immune-related pathways from pathway analysis

To perform ORA, we focused on two Gene Ontology (GO) gene sets: Biological Process (BP) and Molecular Function (MF). Within the BP category, 13 pathways related to immune cell migration and chemokine response were enriched at FDR<0.15 with key proteins from the CXC chemokine ligand family (CXCL2 and CXCL3) and CC chemokine family (CCL19 and CCL23). Among these, 3 pathways ("chemokine-mediated signaling pathway," "response to chemokine," and "cellular response to chemokine") showed enrichment at FDR<0.05 (Fig 3). For MF, 6 pathways associated with chemokine/cytokine activity and immune receptor binding were found to be enriched (FDR<0.15), with 2 pathways ("chemokine activity" and "chemokine receptor binding") achieving FDR<0.05. To perform SPIA, we used the latest version of the Kyoto Encyclopedia of Genes and Genomes (KEGG) pathway database: the only database supported by the tool. Overall, 5 pathways with at least 2 proteins within each pathway were enriched at the FDR<0.15, among which 3 pathways had FDR<0.05. All enriched pathways were cytokine/chemokine-related. Each pathway exhibited downregulation in case individuals compared to controls. S2 Table provides a detailed list of proteins enriched in each pathway.

Notably, many pathways were enriched by the same core sets of highly selected chemokines (CXCL2, CXCL3, CCL19, CCL23), suggesting that these proteins function within interconnected chemokine response processes rather than representing isolated findings. Additional selected proteins, including IL-7, LGALS1, HAVCR2, SELE, and PLVAP, mapped to complementary pathways involving immune exhaustion and regulation, endothelial activation, and leukocyte adhesion, further support the immune signaling network. Moreover, cytokine/chemokine-related pathways remain enriched over a range of threshold selection frequencies (and) used to define the HS protein set. Together, these results demonstrate that our agnostically selected set contains proteins converge on biologically coherent immune pathways that are relevant to breakthrough infections.

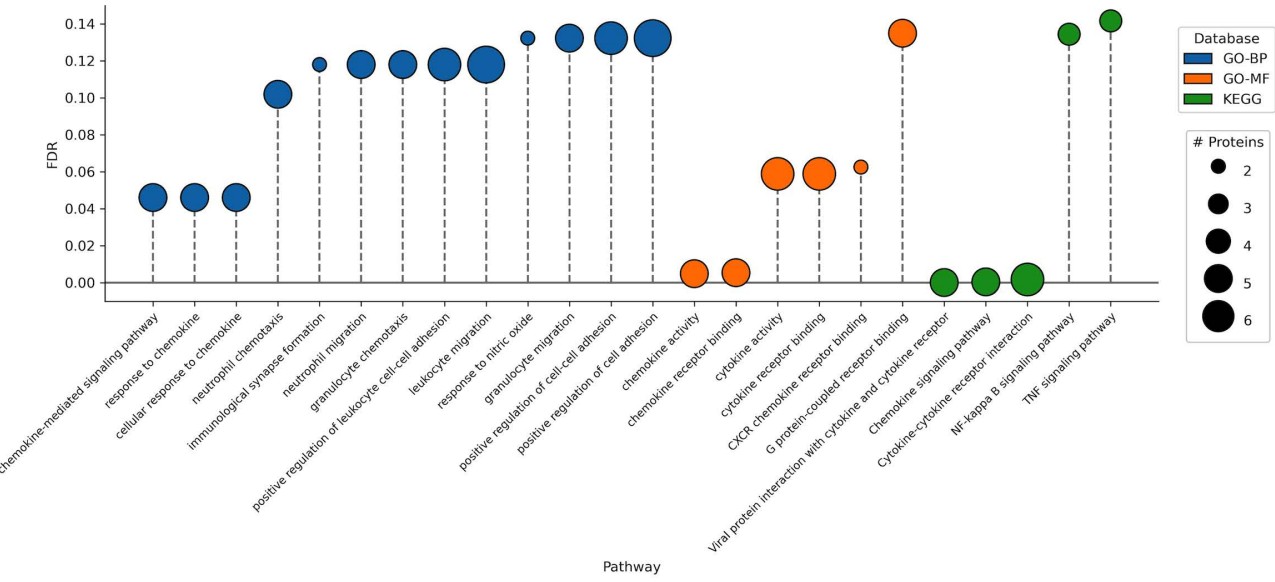

**Fig 3. Dot plot showing pathway enrichment in immune-related pathways.** The pathway dataset is encoded as dot color while number of highly selected proteins found in each pathway is encoded as dot size. Over-representation analysis was performed using the Gene Ontology-Biological Process (GO-BP) and Molecular Function (GO-MF) databases, while signaling pathway impact analysis (SPIA) was performed using KEGG pathways. Though SPIA reports the activation/inhibition of pathways, all shown KEGG pathways were downregulated in case individuals compared with controls.

## Discussion

Our study demonstrated the effectiveness of an elastic-net stability selection approach in reducing a high dimensional dataset of thousands of proteins to a set of tens of proteins consistent with biological understanding. We differentiated this method from conventional univariate or LASSO methods and used both classical overrepresentation analysis and topology-informed pathway enrichment analysis on our set of proteins across three databases, finding enrichment in several immune-related pathways.

Our study highlights the critical role of chemokines in breakthrough infections, which align with findings in the literature [24–29]. Chemokines identified in our study (e.g., CXCL2, CXCL3, CCL19, CCL23) serve as key mediators of the immune system and are known to be essential in recruiting immune cells to sites of infection. We observed that those chemokines were enriched in pathways such as chemokine signaling pathway and cytokine-cytokine receptor interaction (S2 Table). Although chemokines are generally upregulated during viral infections, in our study, they were downregulated in participants with breakthrough infections compared with the control group (Fig 2B). A lower chemokine baseline level after vaccination may suggest susceptibility to Omicron variants by lowering immune readiness in cases vs. controls who are each vaccinated [21,22]. These findings are consistent with the emerging evidence suggesting that Omicron infections are characterized by altered leukocyte trafficking and dysregulated chemokine responsiveness, which could impair immune cell recruitment to sites of viral exposure, potentially allowing Omicron to bypass early immune barriers. Unlike the hyper-inflammatory responses reported in early SARS-CoV-2 infections, our findings may suggest subtler immune vulnerability in vaccinated individuals [30,31]. In addition, immune tolerance mechanisms may also play a role in responding to breakthrough infections, which may suppress the chemokine levels to avoid excessive immune activation.

In contrast, we observed higher levels of IL-7, a cytokine known to maintain T-cell populations and memory cell turnover, in the case group (Fig 2B) [23]. Given that all participants in our cohort had received three doses of the vaccine, this trend lends evidence to immune senescence or exhaustion as a mechanism for decreased immune readiness. Similar observations were made in a previous study which documented increased plasma IL-7 levels in patients with immune failure while undergoing HIV treatment compared to patients with immune success [24]. This hypothesis is further supported by our observation on increased case-group HAVCR2 levels: a canonical marker of NK cell exhaustion [27,32]. Taken together with our chemokine findings, these results suggest a coordinated system with altered immune activation, potential immune exhaustion, and impaired recruitment [33].

Another interesting cytokine, LGALS1, known as galectin-1, also involved in immune responses and inflammation, was observed to have elevated levels in breakthrough cases [26]. Markovic et al. presented results indicating LGALS1 as a significant predictor for COVID-19 severity, which supports its potential role as a staging marker of infection progression [26]. Additionally, E-selectin (SELE) is a recognized marker of endothelial activation, which was shown to have higher levels in ICU-admitted COVID-19 patients, and can potentially serve as a marker for infection monitoring [27]. Notably, chemokine signaling and cytokine-receptor interaction appear to remain core pathways across SARS-CoV-2 variants, suggesting that dysregulation of these immune networks may be a recurring characteristic. Our findings may reflect a shift from the hyperinflammatory immune patterns to a more suppressed or attenuated chemokine response during Omicron infection in vaccinated populations [34,35].

While our analyses generate several provoking hypotheses consistent with published literature, our claims about protein levels between groups and pathway involvement are correlative and require follow-up experiments to establish causality. Additionally, overfitting is a persistent risk in analyzing high dimensional datasets and while our selection of methods was designed to mitigate overfitting, our study would be strengthened with more samples fitting within our matching criteria. Furthermore, proteomic measurements were obtained at a single post-vaccination time point, providing a cross-sectional snapshot of immune status of individuals. The lack of longitudinal data limits the assessment of temporal dynamic of proteomic profiles. Another potential limitation is the healthy worker effect, as the HEROES-RECOVER cohorts consisted of frontline workers who are generally healthier than the broader population. This inherent selection bias

may attenuate observed differences in immune system function and proteomic profiles, thereby tempering the generalizability of our findings to populations with greater comorbidity or frailty. In addition, the near-homogeneous demographic composition (predominantly non-Hispanic White and largely female) limits its generalizability to more diverse populations and potentially other SARS-CoV-2 variants. Although weekly PCR testing irrespective of symptoms and baseline antibody testing for evidence of previous infection minimized the likelihood of undetected prior infection, asymptotic previous infection cannot be ruled out entirely, and unmeasured factors such as hormonal status or medication use may also contribute to proteomic variability. Overall, these considerations highlight the need for larger and more diverse cohorts with longitudinal measurements and external validation to further consolidate the findings.

In conclusion, our analysis provides insights into the immune landscape in breakthrough infections during a unique time period in which it was possible to compare breakthrough infections in a previously infection-naïve population, highlighting the interactions between chemokine signaling and cytokine regulation pathways. These findings emphasize the importance of chemokines in protective immunity to breakthrough infections and hint at immune exhaustion as a mechanism for susceptibility. From a translational perspective, these proteomic signatures may help identify individuals who may remain at elevated risk for infection. With further validation from external independent cohort study, such markers may also help prioritize booster vaccination or guided targeted monitoring strategies. More specifically, several key proteins, such as IL-7, LGALS1, HAVCR2, and SELE may have potential as biomarkers for monitoring and managing infection, offering insights into therapeutic interventions or vaccination schedules. Other proteins were identified in our study, however their roles in breakthrough infections are not well-defined in the context of this investigation and would benefit from further study. Future studies incorporating external validation, larger and more diverse cohorts, and longitudinal samples are essential for further explore these protein markers and pathways to improve our understanding of the mechanisms driving breakthrough infections and inform strategies for disease management.

## Supporting information

**S1 Fig. Histogram of pairwise correlations between log2 RFU of proteins showing mostly moderate to high correlation.**
(DOCX)

**S1 Table. Unpenalized conditional logistic regression model outputs for HS proteins.** From left to right, columns correspond to Entrez gene name, regression coefficients, odds ratio, standard error of coefficients, z-score of coefficients, and p-value of coefficients, respectively.
(DOCX)

**S2 Table. Pathway enrichment significance and membership.** From left to right, columns represent pathway name, p-values computed for the enrichment of each pathway, FDR adjusted p-values, HS proteins contributing to the pathway enrichment, and parent database of each pathway, respectively. P-values for GO-BP and GO-MF database pathways are computed from overrepresentation analysis while KEGG pathways are computed from SPIA (see methods for details).
(DOCX)

**S3 Table. Demographics and health characteristics for frontline workers eligible for the study prior to matching.**
(DOCX)

## Acknowledgments

The authors thank the study participants for their time and commitment, without whom this work would not have been possible. We also thank the reviewers and the editor for their insightful comments, which helped improve the quality of this work.

## Author contributions

**Conceptualization:** Katherine D. Ellingson, Jefferey L. Burgess.

**Data curation:** James Hollister, Tuo Liu, Shawn C. Beitel.

**Formal analysis:** Eric Lu.

**Funding acquisition:** Katherine D. Ellingson, Jefferey L. Burgess.

**Investigation:** Katherine D. Ellingson.

**Methodology:** Yiwen Liu, Eric Lu, Jefferey L. Burgess.

**Project administration:** James Hollister, Shawn C. Beitel.

**Supervision:** Yiwen Liu, Katherine D. Ellingson, Jefferey L. Burgess.

**Visualization:** Yiwen Liu, Eric Lu.

**Writing – original draft:** Yiwen Liu, Eric Lu.

**Writing – review & editing:** Yiwen Liu, Eric Lu, Katherine D. Ellingson, James Hollister, Tuo Liu, Wadana Hamzazai, Shawn C. Beitel, Alberto J. Caban-Martinez, Manjusha Gaglani, Allison L. Naleway, Lauren E.W. Olsho, Andrew L. Phillips, Natasha Schaefer Solle, Harmony L. Tyner, Sarang K. Yoon, Karen Lutrick, Jefferey L. Burgess.

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
