## [Decision Letter · Decision Letter 0]

25 Nov 2025

PONE-D-25-57484Unveiling Post-Vaccination Proteomic Signatures in SARS-CoV-2 Infection-Naïve Individuals Associated with Omicron Breakthrough InfectionsPLOS ONE

Dear Dr. Burgess,

Thank you for submitting your manuscript to PLOS ONE. After careful consideration, we feel that it has merit but does not fully meet PLOS ONE’s publication criteria as it currently stands. Therefore, we invite you to submit a revised version of the manuscript that addresses the points raised during the review process.

We look forward to receiving your revised manuscript.

Kind regards,

Geetika Verma

Academic Editor

PLOS ONE

**Journal Requirements:**

“National Center for Immunization and Respiratory Diseases, Centers for Disease Control and Prevention under contract numbers 75D30120R68013 awarded to Marshfield Clinic Research Laboratory and 75D30120C08379 to University of Arizona.”

3. For studies involving third-party data, we encourage authors to share any data specific to their analyses that they can legally distribute. PLOS recognizes, however, that authors may be using third-party data they do not have the rights to share. When third-party data cannot be publicly shared, authors must provide all information necessary for interested researchers to apply to gain access to the data. (https://journals.plos.org/plosone/s/data-availability#loc-acceptable-data-access-restrictions)

1) A description of the data set and the third-party source.

2) If applicable, verification of permission to use the data set.

3) Confirmation of whether the authors received any special privileges in accessing the data that other researchers would not have.

4) All necessary contact information others would need to apply to gain access to the data.

**Additional Editor Comments:**

The manuscript entitled "Unveiling Post-Vaccination Proteomic Signatures in SARS-CoV-2 Infection-Naïve Individuals Associated with Omicron Breakthrough Infections" has been reviewed and decision for major revision has been made. The authors need to work on the manuscript and incorporate the suggestions made by the reviewers to further enhance the quality of the manuscript.

1. The authors need to provide the acknowledgement section in the manuscript.

2. Please provide the authors' contributions section in the manuscript.

Reviewers' comments:

Reviewer's Responses to Questions

**Comments to the Author**

1. Is the manuscript technically sound, and do the data support the conclusions?

Reviewer #1: Partly

Reviewer #2: Yes

2. Has the statistical analysis been performed appropriately and rigorously?

Reviewer #1: N/A

Reviewer #2: Yes

3. Have the authors made all data underlying the findings in their manuscript fully available?

Reviewer #1: Yes

Reviewer #2: Yes

4. Is the manuscript presented in an intelligible fashion and written in standard English?

Reviewer #1: Yes

Reviewer #2: Yes

5. Review Comments to the Author

Reviewer #1: This manuscript presents a timely and technically sound study examining proteomic profiles in vaccinated, infection-naïve individuals who later experienced Omicron breakthrough infections. The authors identify key protein signatures and immune pathways associated with susceptibility to infection. The topic is highly relevant for understanding correlates of protection in the post-vaccination era, and the dataset appears robust. The study is clearly written and well-organized, but several sections would benefit from additional detail, clarification, and contextual discussion to make the findings more interpretable and broadly impactful.

Comments

1. The introduction provides good background on vaccine immunity and breakthrough infections, but it could more explicitly highlight what unique insight proteomic profiling adds beyond antibody and T-cell studies. Clarifying the central hypothesis why proteomics, and what new biology it reveals would make the motivation stronger.

2. The paper would benefit from a more detailed description of the cohort age, sex, vaccination type, timing relative to infection, and infection ascertainment method. It’s also important to specify how breakthrough and control subjects were matched and whether potential confounders were adjusted for in the analysis.

3. The proteomic platform (e.g., SOMAscan, Olink, LC–MS/MS) should be specified, along with normalization procedures, quality control measures, and correction for batch effects. Likewise, the rationale for using unpenalized conditional logistic regression should be briefly explained, along with how multiple testing was handled across thousands of proteins.

4. The results table lists significant associations (e.g., LGALS1, PDLIM4, VAMP2, HAVCR2, SELE, CCL19), but the text could elaborate on how these proteins fit together functionally. Are they connected within common chemokine or leukocyte signaling pathways, or do they represent independent immune processes? Providing a concise pathway-level summary would help readers understand the biological implications.

5. Pathway analysis is strong and consistent, highlighting chemokine-mediated signaling and cytokine-receptor interactions. Still, the discussion could better connect these findings to known immune mechanisms during Omicron infection such as altered leukocyte trafficking, hyperinflammatory responses, or immune evasion. It would also be valuable to mention whether similar proteomic signatures have been observed in earlier SARS-CoV-2 variants.

6. The discussion could emphasize how the identified proteomic markers might be used in practice for example, as predictive biomarkers for breakthrough risk, or to guide booster vaccination strategies. Including a short “limitations and future directions” paragraph (e.g., small cohort size, timing of sampling, or generalizability to other variants) would also strengthen the conclusion.

7. The abstract should mention at least one or two quantitative findings (e.g., number of significant proteins or key enriched pathways).

8. Add recent references (2023–2024) on post-vaccination systems-level analyses for better context.

Reviewer #2: This study exhibits high scientific rigor, with careful participant matching, robust statistical methodology (EN-CLR with bootstrap stability selection), and thoughtful pathway-level validation. The analytical design successfully addresses the pitfalls of high-dimensional proteomic data and avoids overreliance on single-model feature selection. The identification of chemokine and cytokine alterations linked to breakthrough infections provides biologically coherent insights consistent with current immunological understanding.

However, few limitations merit consideration.

1. The near-homogeneous population (97% non-Hispanic white, 74% female) and inclusion of primarily healthy frontline workers introduce a substantial healthy worker bias and limit generalizability to more diverse or comorbid populations.

2. The single post-vaccination sampling time (≈4 weeks after dose three) provides only a snapshot of immune status, precluding conclusions about temporal dynamics of protein expression or causal directionality.

3. Despite using stability selection, the sample-to-variable ratio remains low (~360 participants vs. >7,000 proteins), increasing the risk of overfitting and spurious associations. Independent validation in an external cohort or replication dataset would greatly strengthen confidence.

4. The mechanistic inference that downregulated chemokines indicate immune exhaustion is speculative without corroborating cellular or longitudinal data. Functional assays (e.g., T-cell activation, cytokine release) are needed to validate proteomic signatures biologically.

5. Many enriched GO and KEGG terms overlap heavily in chemokine-related functions, raising the possibility that apparent enrichment reflects pathway redundancy rather than distinct biological processes.

6. Variables such as prior asymptomatic infection, hormonal status, or medication use are not described but could influence circulating protein levels. Their omission may partially confound observed associations.

7. The manuscript would benefit from explicit mention of cross-validation parameters, stability thresholds (why S > 500 was chosen), and sensitivity analyses for model robustness.

Overall, while the work is methodologically innovative and timely, its translational relevance would be greatly enhanced by validation in more heterogeneous populations, longitudinal sampling, and orthogonal functional assays to confirm causality and biomarker potential. These areas require clarification and improvement before it can be accepted for publication.

6. PLOS authors have the option to publish the peer review history of their article (what does this mean?). If published, this will include your full peer review and any attached files.

Reviewer #1: **Yes:** Dr. Prashant Singh

Reviewer #2: No

---

## [Author Response · Author response to Decision Letter 1]

2 Feb 2026

Please see the separate response to reviewers document.

---

## [Decision Letter · Decision Letter 1]

6 Apr 2026

Unveiling Post-Vaccination Proteomic Signatures in SARS-CoV-2 Infection-Naïve Individuals Associated with Omicron Breakthrough Infections

PONE-D-25-57484R1

Dear Dr. Burgess,

We’re pleased to inform you that your manuscript has been judged scientifically suitable for publication and will be formally accepted for publication once it meets all outstanding technical requirements.

Kind regards,

Sawar Khan, Ph.D

Academic Editor

PLOS One

Additional Editor Comments (optional):

Reviewers' comments:

Reviewer's Responses to Questions

**Comments to the Author**

1. If the authors have adequately addressed your comments raised in a previous round of review and you feel that this manuscript is now acceptable for publication, you may indicate that here to bypass the “Comments to the Author” section, enter your conflict of interest statement in the “Confidential to Editor” section, and submit your "Accept" recommendation.

Reviewer #1: All comments have been addressed

Reviewer #2: (No Response)

Reviewer #3: All comments have been addressed

2. Is the manuscript technically sound, and do the data support the conclusions?

Reviewer #1: Yes

Reviewer #2: Yes

Reviewer #3: Yes

3. Has the statistical analysis been performed appropriately and rigorously?

Reviewer #1: N/A

Reviewer #2: Yes

Reviewer #3: Yes

4. Have the authors made all data underlying the findings in their manuscript fully available?

Reviewer #1: Yes

Reviewer #2: No

Reviewer #3: Yes

5. Is the manuscript presented in an intelligible fashion and written in standard English?

Reviewer #1: Yes

Reviewer #2: Yes

Reviewer #3: Yes

6. Review Comments to the Author

Reviewer #1: The authors have done an excellent job revising the manuscript. They have addressed all my comments thoroughly and efficiently. The manuscript is now ready for publication, and I congratulate the authors on a well-executed and impactful study.

Reviewer #2: This manuscript presents a nested case- control proteomic analysis within the HEROES- RECOVER frontline worker cohorts to identify baseline serum protein signatures associated with subsequent Omicron breakthrough infections in individuals who received three doses of WA-1 mRNA vaccines. Using high-dimensional SomaScan proteomics (>7,000 proteins) combined with elastic-net regularized conditional logistic regression and bootstrapped stability selection, the authors identified 28 highly selected proteins distinguishing breakthrough cases from matched controls. Key findings include reduced baseline chemokines (CXCL2, CXCL3, CCL19, CCL23), increased IL-7 levels, and enrichment of chemokine/cytokine signaling pathways. The study proposes that altered immune readiness and compensatory T-cell regeneration may contribute to susceptibility to breakthrough infection.

The manuscript is timely and methodologically sophisticated, with strengths in cohort design, matching strategy, and advanced statistical modeling. However, several areas require clarification, including mechanistic interpretation, validation strategy, generalizability, and potential residual confounding.

Major Comments

1. The manuscript risks overinterpreting baseline proteomic differences as mechanistic drivers of susceptibility. Single pre-infection timepoint prevents assessment of dynamic immune trajectories. Elevated IL-7 and exhaustion markers are interpreted as immune senescence without functional validation. Authors may explicitly frame findings as predictive associations rather than causal mechanisms. Discuss alternative explanations (e.g., immune homeostasis, recent antigen exposure, subclinical infection) and include sensitivity analyses stratified by time between blood draw and infection.

2. The choice of selection frequency S>500 appears arbitrary. Authors should provide rationale or simulation-based justification for the cutoff, include stability curves showing protein selection frequencies and may consider reporting false discovery control metrics or selection probability thresholds.

3. Despite matching, important variables may influence proteomic signatures: Occupational exposure intensity, hormonal status, medication use (e.g., corticosteroids) and time since last vaccine antigen exposure. Authors may perform adjusted conditional logistic regression including additional covariates where possible.

4. Cohort is predominantly non-Hispanic White and female frontline workers. Findings may not translate to elderly, immunocompromised, or globally diverse populations. Authors may emphasize limited external generalizability in discussion by including demographic representativeness comparison with broader vaccinated populations.

5. SomaScan measures aptamer-based relative abundance rather than absolute protein concentration. Potential cross-reactivity or isoform ambiguity should be acknowledged. Authors may discuss validation requirements using orthogonal platforms (e.g., ELISA, mass spectrometry) and clarify how multiple SOMAmers mapping to the same protein were handled.

Minor Comments

• Clarify whether samples were collected fasting or at standardized times.

• Provide distribution plots or PCA demonstrating batch correction effectiveness.

• Improve wording around “immune exhaustion” to avoid overinterpretation.

• Include effect sizes with confidence intervals for highly selected proteins.

• Consider visual network diagrams illustrating chemokine-centered pathways.

Reviewer #3: The manuscript entitled “Unveiling Post-Vaccination Proteomic Signatures in SARS-CoV-2 Infection-Naïve Individuals Associated with Omicron Breakthrough Infections” has been carefully revised by the authors. All comments and suggestions raised during the review process have been adequately addressed. The revisions have improved the clarity, quality, and overall presentation of the manuscript. The study is well-structured, and the findings are clearly presented. The manuscript is now suitable for publication.

7. PLOS authors have the option to publish the peer review history of their article (what does this mean?). If published, this will include your full peer review and any attached files.

Reviewer #1: **Yes:** Prashant Singh

Reviewer #2: **Yes:** Naseem akhter

Reviewer #3: No

---

## [Editor Report · Acceptance letter]

PONE-D-25-57484R1

PLOS One

Dear Dr. Burgess,

I'm pleased to inform you that your manuscript has been deemed suitable for publication in PLOS One. Congratulations! Your manuscript is now being handed over to our production team.

Kind regards,

on behalf of

Dr. Sawar Khan

Academic Editor

PLOS One